# RGFP966 inhibits activation of AIM2 inflammasomes to promote mitophagy to relieve acute gouty arthritis

Zewei Wang[1], Chunyang Liu[1], Feng Cheng[1], Xiuyun Shi[1], Jing Dai[1], Jing Tian[2]*, Hongxu Jin[1]*, Ying Liu[1]*

1 Laboratory of Rescue Center of Severe Wound and Trauma PLA, Emergency Medicine Department, General Hospital of Northern Theater Command, Shenyang, Liaoning, China, 2 Department of Orthopedics, General Hospital of Northern Theater Command, Shenyang, Liaoning, China

* 13352459336@163.com (JT); cszx_jhx@163.com (HJ); cszx_ly@163.com (YL)

## Abstract

Acute gouty arthritis (AGA) is a common inflammatory joint disease characterized by pain resulting from the deposition of monosodium urate (MSU) crystals into joints and surrounding tissues. RGFP966, a selective inhibitor of histone deacetylases 3 (HDAC3), can down-regulate the AIM2 inflammasomes. This study aimed to explore the mechanism of action of RGFP966 in MSU-induced AGA. MSU-induced AGA rats were treated with RGFP966 or colchicine, and physiological and pathological indicators were determined. The rat joint synovial tissues and kidneys pathological damage was observed by hematoxylin and eosin staining. Enzyme-linked immunosorbent assay (ELISA), Western blot, JC-1 staining, mitochondrial membrane potential (MMP) measurement, ATP assay, and immunofluorescence were applied to evaluate the effects of RGFP966 on the AIM2 inflammasomes and mitophagy. In MSU-induced AGA rats, RGFP966 significantly improved the gait score, the swelling degree and synovitis score. In addition, RGFP966 repressed the increased serum levels of IL-1β, IL-18, IL-6 and TNF-α, and reduced the protein levels of AIM2, Pro-caspase-1, Cleaved-caspase-1, ASC, Pro-IL-1β, and Cleaved-IL-1β in MSU-induced AGA rats. Furthermore, RGFP966 decreased the level of reactive oxygen species (ROS), increased the level of ATP and MMP, and promoted the levels of Pink1, Parkin and LC3-II. Collectively, RGFP966 significantly alleviated AGA, and the underlying mechanism is related to promote mitophagy by inhibiting the activation of the AIM2 inflammasomes.

## Introduction

Acute gouty arthritis (AGA) is a joint inflammation caused by a disorder of purine metabolism and impaired excretion of uric acid, which lead to elevated uric acid levels in the body [1]. AGA causes the deposition of monosodium urate (MSU)

**Data availability statement:** All data are in the manuscript and/or Supporting information files.

**Funding:** The author(s) declare financial support was received for the research, authorship, and/or publication of this article. This study was financed by the Natural Science Foundation of Liaoning Province (2023-MSLH-361). The project funder in this study is Dr. Jing Tian, who is mainly responsible for "Funding acquisition, supervision, and Writing-review & editing".

**Competing interests:** The authors have declared that no competing interests exist.

crystals into the joints and surrounding tissues [2]. The clinical manifestations of AGA are recurrent episodes of joint pain, redness, swelling, and even joint dysfunction [3]. The incidence of AGA is rapidly increasing in China. Without proper treatment, AGA may develop into a chronic inflammatory disease and lead to physical disability and renal impairment, which seriously affects patient quality of life [4,5]. Inflammasomes, composed of pattern recognition receptors (PRRs), apoptosis-associated speck-like protein containing an apoptosis-associated speck-like protein (ASC), and inactive pro-caspase-1, are multiprotein complexes found in the cytoplasm [6]. They can only function after they are activated. The PRRs of inflammasomes mainly include members of the nucleotide-binding oligomerization domain-like receptor (NLR) family and the absent in melanoma 2 (AIM2)-like receptor (ALR) family [7]. Although many literatures have reported that AGA is associated with the activation of NLRP3, research on AIM2 remains limited. MSU crystals can recruit and activate neutrophils, and releasing neutrophil extracellular traps (NETs) is key factor in the pathogenesis of AGA [8,9]. NETs are complex net structures of multiple proteins and DNA that are released under pathological conditions to recognize DNA receptors to exert strong pro-inflammatory effects [10]. AIM2 is an intracellular inflammasome that recognizes cytoplasmic DNA [11]. AIM2 inflammasomes are protein complexes composed of AIM2, ASC, and Caspase-1 [12]. It has been found that MSU crystals release NETs when AGA occurs [13]. The main component of NETs, DNA, combines to the AIM2 inflammasomes in the cytoplasm of synovial cells in the joints, activating Caspase-1, shearing GSDMD to induce cellular pyroptosis, and then releasing more IL-1β, and IL-18 to amplify the inflammatory response [14,15].

RGFP966 (C21H19FN4O) is a selective inhibitor of histone deacetylase 3 (HDAC3) that mitigates ischaemic brain injury by downregulating AIM2 inflammasome expression [16]. RGFP966 may inhibit AIM2 by regulating the acetylation and phosphorylation of signal transducer and activator of transcription 1 (STAT1). Acetylation of STAT1 inhibits its phosphorylation, and RGFP966 suppresses its activation by directly promoting STAT1 acetylation, thereby inhibiting the pathway leading to AIM2 inflammasome activation [17]. One study also showed that RGFP966 reduced the clinical indications of ankle joint injury and alleviated cartilage and bone destruction in adjuvant arthritis (AA) rats [18]. It has been shown that activation of the AIM2 inflammasomes lead to Caspase-1-dependent mitochondrial damage and massive reactive oxygen species (ROS) production [19]. Autophagy is turned on as a protective mechanism when cells undergo damage caused by oxidative stress [20], hypoxia [21], trauma and other environmental factors [22,23]. Mitophagy is a selective autophagy that removes damaged, senescent or redundant mitochondria [24]. It is also an important regulatory mechanism for mitochondrial quality control, which plays an important role in maintaining cellular homeostasis [25]. Mitochondrial morphology and function are markedly altered in osteoarthritis, leading to pathologic changes such as chondrocyte apoptosis and extracellular matrix degradation, and mitophagy may serve as a target for the treatment of AGA [26]. The Pink1/Parkin

signaling pathway is a key initiator of mitophagy [27]. Nevertheless, it is still unclear whether RGFP966 regulates mitophagy by inhibiting activation of AIM2 inflammasomes to relieve AGA.

In order to reveal the role and mechanism of RGFP966-mediated AIM2 inflammasomes in the regulation of the Pink1/Parkin signaling pathway, we established an AGA rat model and used colchicine as the positive control. We investigated the effects of RGFP966, a potential AIM2 inhibitor, on inflammatory and mitophagy-related markers in rat synovial tissues, thereby providing new insights into potential therapeutic targets for the prevention and treatment of gout.

## Materials and methods

### Animals

6-week-old male Sprague-Dawley rats (200±20g) were obtained from Liaoning Changsheng Laboratory Animal Co. (License: SCXK [Liao J2020-0001]). Before the experiment, the rats were kept in separate cages for one week at room temperature, and were allowed to consume food and water freely. The animal study was reviewed and approved by Medical Ethics Committee of the General Hospital of Northern Theater Command. (Ethical approval number: 2024−63).

### Experiment design

The experimental rats were randomly divided into four groups (n=24): (1) Control group, (2) MSU group, (3) MSU+RGFP966 group, and (4) MSU+Colchicine group. According to the conversion ratio of human and animal body surface area, the colchicine group was gavaged with 0.3 mg/kg dose of colchicine (HY-16569; MedChemExpress) solution once a day for 7 consecutive days as a positive control. RGFP966 (S28100-5 mg; MedMol), a specific inhibitor of HDAC3, was dissolved in 5% DMSO to form a reserve solution of 20 µM and stored at −80°C, then added to corn oil to form a working solution, mixed well. 50 µL of RGFP966 was injected into the joint cavity of RGFP966 group from day 5 according to a previously published paper [18], and an equal amount of corn oil containing 5% DMSO was injected into the joint cavities of the control group and the MSU group once daily for 3 consecutive days. The model of AGA was established on the 6th day, and the modeling was followed by RGFP966 or colchicine treatment for 1 day. 100 mg of MSU (U2875; Sigma) crystal was dissolved in 3.6mL of saline and 0.4mL of Tween 80 (T8360, Solarbio). After heating and stirring for 5 min, the solution was ultrasonic crushed for 10 min, and prepared into 4mL MSU white suspension. 200 µL of MSU suspension was injected into the joint cavity of the knee joint of rats in the MSU group, RGFP966 group and colchicine group, and the same dose of saline was injected into the joint cavity of the knee joint of rats in the control group [28]. If the joints were swollen and the joint movements were limited, it indicated that the AGA model was successfully prepared. For the surgery, rats were anesthetized with an intraperitoneal injection of sodium pentobarbital (40 mg/kg). Efforts to alleviate pain and suffering include providing appropriate analgesia and anesthesia for the duration of the experiment, as well as ensuring adequate housing conditions for the animals. After anesthetizing the animal, the abdominal aorta separates and blood is quickly drawn out, and the animal dies, followed by follow-up experimental testing. The synovial tissues of the knee joints and kidneys were removed on the 8th day. The methods for animal model construction are shown in Fig 1. The mental state, feeding, flexibility, and body weight of the rats before modeling and after modeling treatment were observed and recorded.

### Measurement of joint swelling degree and gait scores

The circumferential diameters of the left knee joint were measured at 6, 12, 24, and 48 h before and after modeling. To ensure the accuracy of the measurements, the whole procedure was done by the same person, and the measurements were repeated 3 times in each group to take the average value. The specific calculation formula of the joint swelling rate was delineated below: The joint swelling rate (%) = (joint circumference at the time of measurement −initial circumference)/initial circumference ×100%. The Coderre gait grading standard was used to score the gait of rats at 6,12, 24, and 48 h [29].

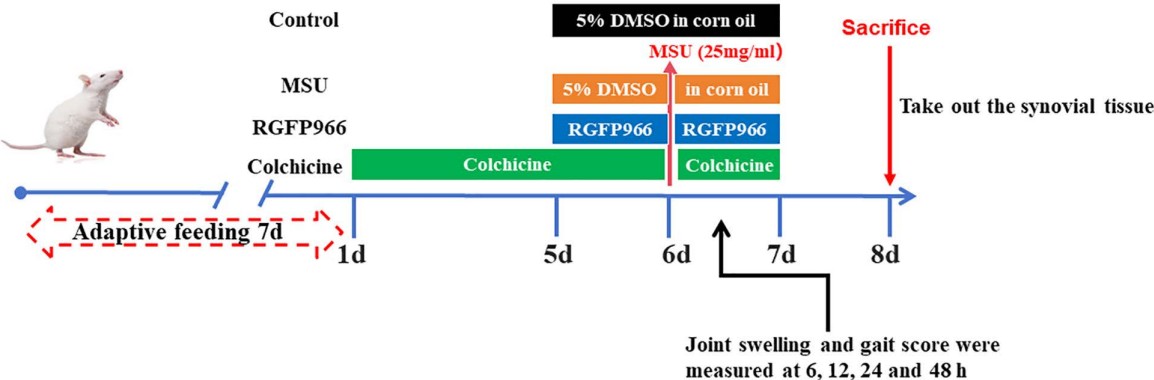

**Fig 1. The methods for animal model construction are shown.**

## Hematoxylin and eosin (H&E) staining

Knee synovial and kidney tissues were immersed in 4% paraformaldehyde and fixed for 48 h. The specimens were embedded in paraffin after dehydration and cut into about 5 μm thick sections for histological analysis. The sections were morphologically evaluated according to the H&E staining kit (G1120; Solarbio). Briefly, the sections were stained with hematoxylin staining solution for 10 min, differentiated by differentiation solution for 30 s, and stained with eosin staining solution for 1 min. Subsequently, the histopathological evaluations of synovial tissues were carried out [5,30].

## Mitochondrial extraction

The mitochondria of the synovial tissues were extracted by Animal mitochondrial extraction kit (HR0233; Biolab). Briefly, 50 mg of synovial tissue was collected, and cleaned with pre-cooled PBS and cut into pieces. The tissue was added with pre-cooled mitochondrial extraction solution, homogenized and centrifuged for several times (11000 × g, 20 min, 4°C). Subsequently, the supernatant was discarded, and the precipitate was the mitochondria, which was resuspend by adding appropriate amount of mitochondrial storage solution.

## Mitochondrial membrane potential (MMP) measurement

The MMP levels were measured by Mitochondrial membrane potential assay kit (C2006; Beyotime) according to the manufacturer's instructions. In brief, JC-1 staining working solution was prepared and the concentration of mitochondria in synovial tissue by BCA was determined. The prepared JC-1 staining solution was diluted 5 times, and then the diluted staining solution and the purified mitochondria of synovial tissue were added in the ratio of 9:1. The fluorescence values of aggregate JC-1 (excitation and emission wavelengths of 585 and 590 nm, respectively) and monomer JC-1 (excitation and emission wavelengths of 514 and 529 nm, respectively) were measured in a fluorescent enzyme marker.

## Western blot

The synovial tissues of the left knee joint of rats were taken, fully sheared, lysed by RIPA buffer solution and fully homogenized, centrifuged at ultra-low temperature (4 °C, 12000 r/min, 20 min/times, a total of 2 times), and the supernatant was extracted and set aside. The OD value of the BCA protein kit (WB6501, New Cell & Molecular Biotech) quantitatively was measured at n = 562 nm, and the concentration of the sample proteins and the amount of protein on the sample were calculated. The proteins were denatured and subjected to SDS-PAGE electrophoresis, wet transferred to PVDF membrane, and closed at room temperature with milk containment solution for 2 h. The PVDF membrane was washed three times

with TBS-T for 5 min/times, and prepared with β-actin (1:1000), AIM2 (1:1000, YP-Ab-03685, UpingBio), Pro-caspase-1 (1:1000, AF5418, Affinity), Cleaved-caspase-1 (1:1000, AF4005, Affinity), Pro-IL-1β (1:1000, AF5103, Affinity), Cleaved-IL-1β (1:1000, AF4006, Affinity) and PINK1 (1:1000, DF7742, Affinity), Parkin (1:1000, YP-Ab-12790, UpingBio), and LC3 (1:1000, AF5402, Affinity) antibodies at 4°C overnight. The next day, the PVDF membrane was removed at room temperature and washed three times in TBS-T for 5 min/time, and goat anti-rabbit IgG-HRP secondary antibody (1:5000) was incubated at room temperature for 2 h. The protein bands were visualized with an ECL kit (P10060; New Cell & Molecular Biotech) by a chemiluminescence detection system.

### ELISA

Blood was collected from the abdominal aorta of each group of rats and serum was separated. The levels of IL-1β, IL-18, IL-6 and TNF-α in serum and ROS in the synovial tissues were analyzed by ELISA kits (YJ696735, YJ35345, YJ828433, YJ002095, YJ998771; Shanghai Meilian Biotechnology).

### ATP assay

ATP contents of the synovial tissues were detected by ATP assay kit (A095-1–1; Nanjing Jiancheng Bioengineering Institute) according to the manufacturer's instructions. Briefly, the synovial tissues were accurately weighed and added 9 times the volume of pre-cooled double-distilled water. The supernatant was taken for measurement after homogenized and centrifuged. Blank, standard, assay and control tubes were prepared according to the instructions, and the OD value of each tube was measured at 636 nm. Subsequently, ATP content in the synovial tissues of each group was calculated by using the calculation formula.

### Immunofluorescence

The collected synovial tissues were fixed, embedded and sectioned. Knee synovial tissues were boiled in Sodium Citrate Antigen Retrieval Solution (C1032, Solarbio) for 10 min and then exposed to 0.2% Triton X-100 (T8200, Solarbio) for 15 min for permeation, and closed with 5% BSA for 30 min to exclude nonspecific staining. Subsequently, the sections were incubated with the primary antibodies at 4°C overnight. For double immunofluorescence staining, AIM2 antibody (1:200, sc-293174, Santa Cruz Biotechnology) and Caspase-1 antibody (1:300, AF5418, Affinity) were added to the sections in the wet box was kept at 4 °C overnight. On the next day, the synovial sections were washed by PBST and then secondary antibodies Goat anti-mouse IgG H&L (Alexa Fluor 647) (1:500, ab150115, Abcam) and Goat anti-rabbit IgG H&L (Alexa Fluor 488) (1:500, ab150071, Abcam) mixed were incubated for 2 h at room temperature. DAPI was exploited for nuclear staining for 10 min. Finally, the slides were sealed with Mounting Medium, antifading (S2100, Solarbio), and images were captured under a fluorescence microscope.

### Statistical analysis

All data were performed after three and more independent and repeated tests. Statistical analysis were performed by using Prism 10.0 and all data were expressed as mean±standard deviation. The independent sample t-test was used for comparison between the two groups, and data were analyzed using One-way ANVOA or Two-way ANVOA among multiple groups. The difference was statistically significant when $P < .05$.

## Results

### Observations on the general condition of rats after modeling

The rats in each group had glossy hair color, normal knee gait, rapid response, and normal food and water intake before modeling. After injection of MSU crystals, the knee joints of rats showed obvious swelling at 6 h. The swelling degree

of rats with drug intervention was less severe than that of the MSU group at 24 and 48 h (Fig 2A). The swelling degree of knee joints injected in MSU crystals all reached the highest point at 6 h and was significant increased compared with that of the control group rats ($P < 0.0001$). The rats in the RGFP966 and colchicine groups showed a significant decrease in swelling degree from 6 to 48 h. No statistically significant difference was oberved between the two drug intervention groups ($P > 0.05$) (Fig 2B). Observing the initial body weight of rats in each group and the change of body weight after modeling treatment, compared with the initial body weight, the rats in the colchicine group were suffered from diarrhea and weight loss, whereas the body weight of rats increased significantly in all other groups (Fig 2C).

## RGFP966 Alleviated AGA in Rats

Compared with the control group, rats in the MSU group showed different degrees of lameness at 6, 12, 24 and 48 h after modeling, with statistically significant differences in gait scores at all time points. At 6, 12, 24 and 48 h after MSU induction, both RGFP966 and colchhicine improved the gait score of rats (all $P < 0.05$) (Fig 3A). Moreover, after HE staining

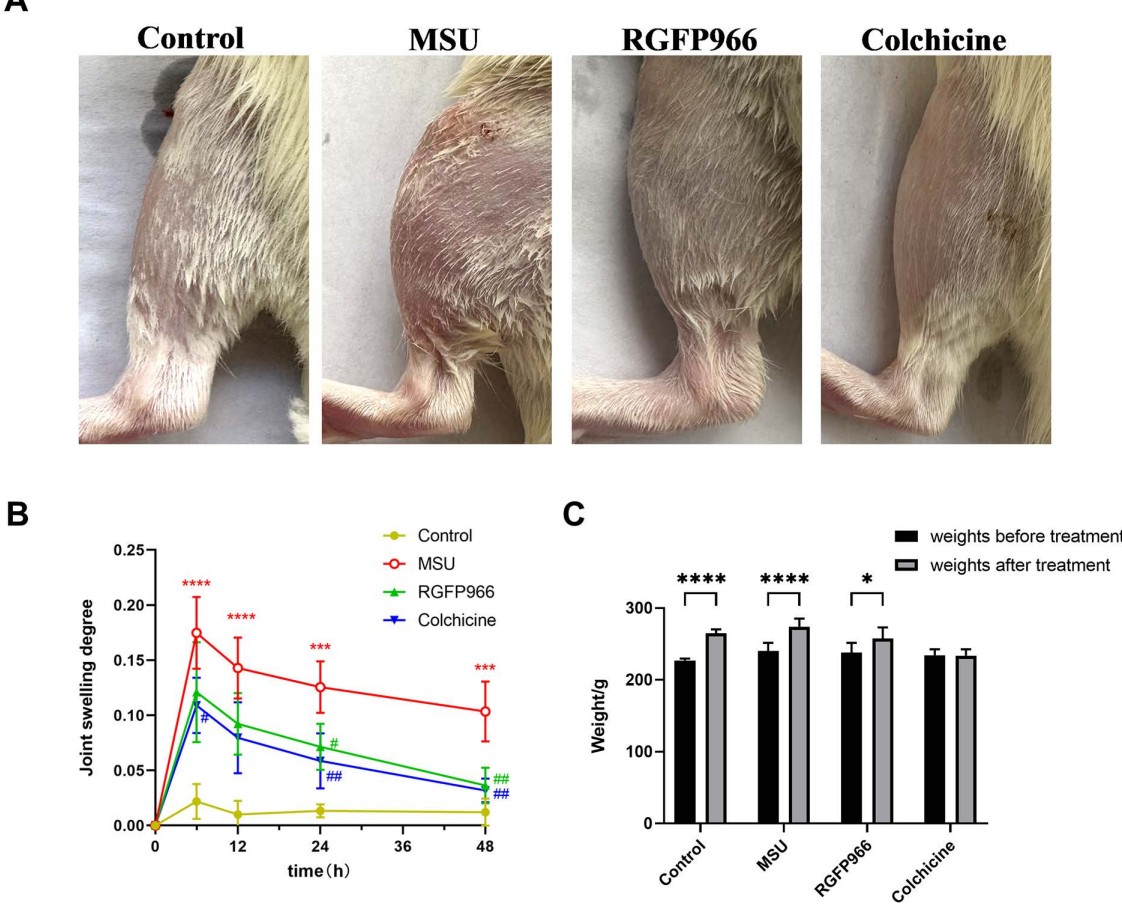

**Fig 2. Observations on the general condition of rats after modeling.** Representative images of the knee joints of rats in each group at the 6th hour after MSU injection (A). Joint swelling degree of rats in the Control, MSU, MSU+RGFP966 and MSU+Colchicine groups was evaluated 6, 12, 24, and 48 h after MSU injection (B). ***$P < 0.001$, ****$P < 0.0001$ verus Control group; #$P < 0.05$, ##$P < 0.01$ verus MSU group. Changes in body weight before and after treatment in each group of rats (C). ***$P < 0.001$, ****$P < 0.0001$ verus weights before treatment. The data was presented as means ± standard deviation. n = 6/group.

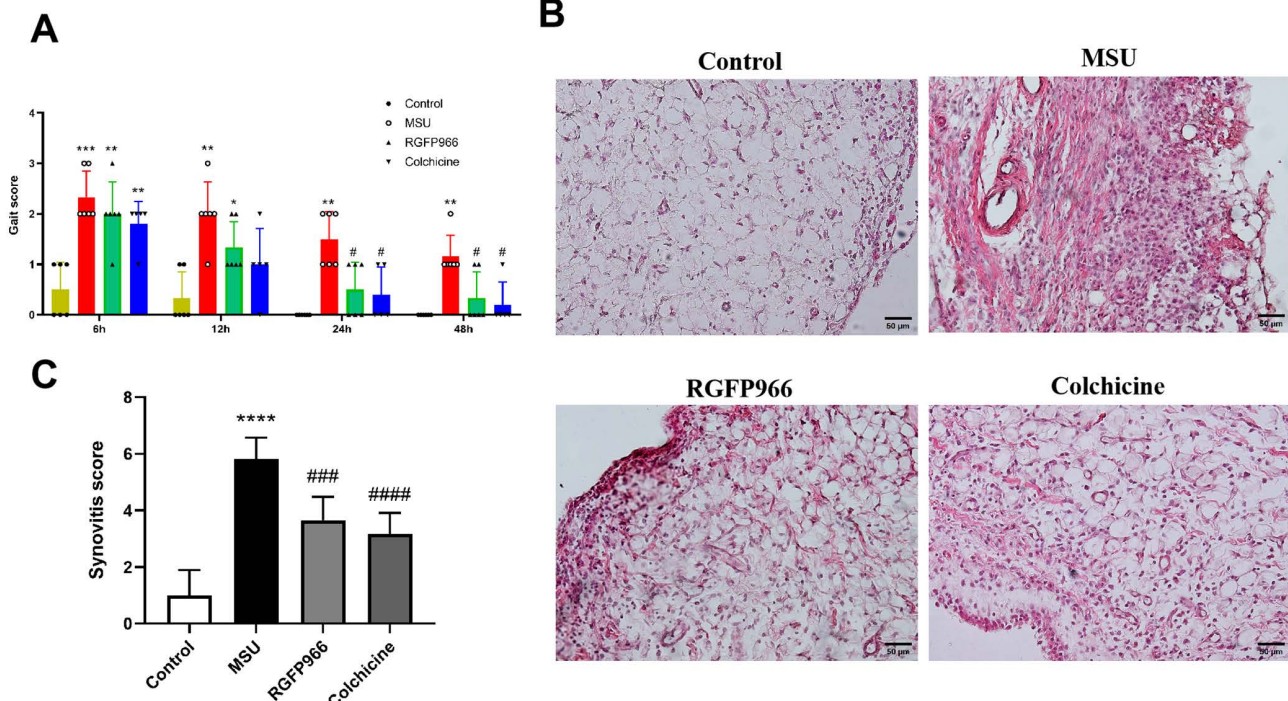

**Fig 3. RGFP966 ameliorated inflammation in the MSU-induced AGA rat model.** After joint cavity injection of 0.2 mL of 25 mg/mL MSU solution on day 6, gait scoring was performed at 6, 12, 24 and 48 h (A). Histopathologic changes in rat synovial tissues observed by H&E staining (×200, Scale bar = 50 μm) (B) and synovial histopathology is measured by semi-quantitative score (C). $^*P < 0.05$, $^{**}P < 0.01$, $^{***}P < 0.001$, $^{****}P < 0.0001$ verus Control group; $^#P < 0.05$, $^{##}P < 0.01$, $^{###}P < 0.001$, $^{####}P < 0.0001$ verus MSU group. The data was presented as means ± standard deviation. n = 6/group.

of the synovium of the joint, the synovial cells in the lining layer of rats in the control group were arranged normally, with clear boundaries, and no obvious inflammatory cell infiltration was seen. The synovial tissue of rats was structurally disorganized, with unclear boundaries, obvious synovial cells hyperplasia, a large number of inflammatory cell infiltration, and vascular hyperplasia after MSU treatment. Meanwhile, the synovitis score increased notably ($P < 0.0001$). RGFP966 and colchicine ameliorated MSU-induced histopathological injury, synovial cells hyperplasia and inflammatory cells infiltration (all $P < 0.001$) (Fig 3B and C).

### Effects of RGFP966 on inflammatory cytokines levels in the serum of rats

To further verify that RGFP966 could improve the inflammation level in AGA rats, ELISA results showed that the serum levels of IL-1β, IL-18, IL-6, and TNF-α were significantly increased in rats in the MSU group. It suggested that the inflammation in the knee joints of the MSU-induced AGA rats was aggravated and the modeling was successful. The serum levels of IL-1β, IL-18, IL-6, and TNF-α in AGA rats were significantly decreased after RGFP966 and colchicine treatment, but there was no significant difference between the serum levels of inflammatory factors in the two treatment groups (Fig 4).

### RGFP966 decreased the AIM2 inflammasomes expression in AGA rats

To further investigate the regulation of AIM2 inflammasomes in AGA rats, we examined the expression of the AIM2 inflammasome complexes and inflammatory cytokines by Western blot and immunofluorescence staining. The production of the AIM2 inflammasomes and secretion of IL-1β in the synovial tissues were activated by MSU.

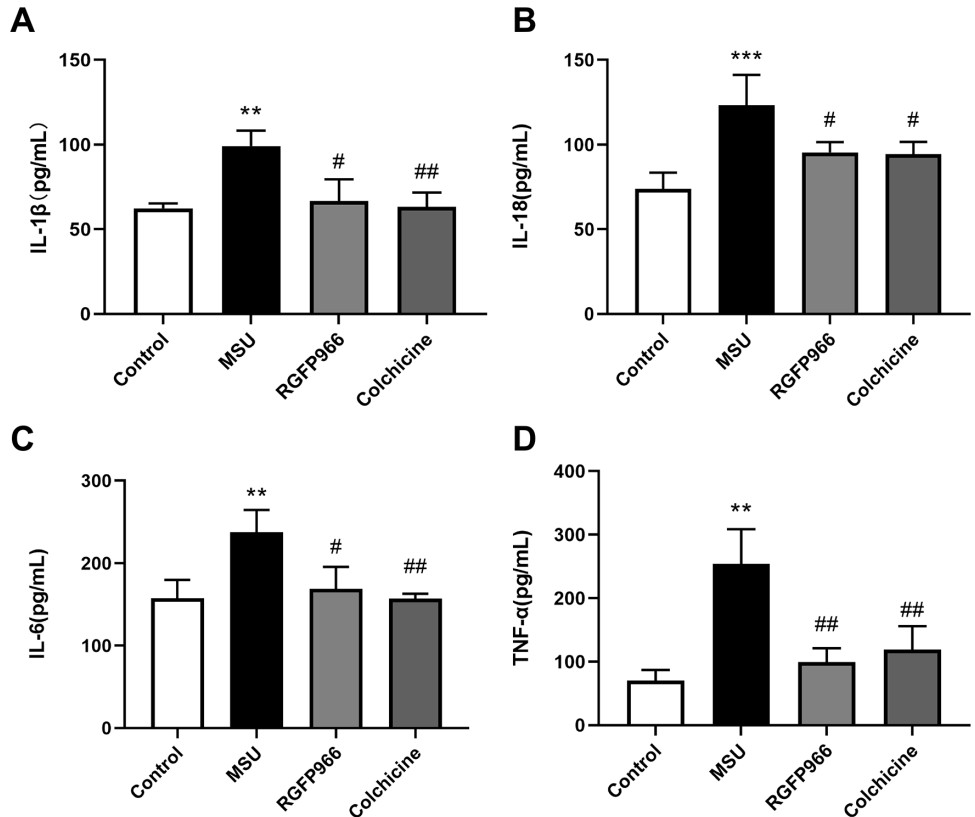

**Fig 4. The levels of IL-1β (A), IL-18 (B), IL-6 (C), and TNF-α (D) in rat serum were detected by ELISA.** $^{**}P<0.01$, $^{***}P<0.001$ verus Control group; $^{\#}P<0.05$, $^{\#\#}P<0.01$ verus MSU group. The data was presented as means ± standard deviation.

The expression of AIM2, ASC, Pro-caspase-1, Cleaved-caspase-1, Pro-IL-1β and Cleaved-IL-1β in the synovial tissues of the knee joints of the rats in the MSU group were significantly increased compared with the control group, whereas the expression of the above proteins in the synovial tissues of the knee joints of the rats after RGFP966 and colchicine treatment were significantly decreased (Fig 5A-G). MSU treatment promoted the contents of AIM2 and caspase-1 in the synovial tissues after RGFP966 treatment, which was detected by immunofluorescence (Fig 5H-J). It suggested that RGFP966 could block the activation of the AIM2 inflammasomes in AGA rats.

## RGFP966 decreased mitochondrial damage in an experimental rat model of AGA

It has been reported that the inflammasomes activation to be associated with defective mitochondrial function and ROS accumulation [31]. Hence, to observe the mitochondrial function in the synovial tissues, we further examined ROS levels, ATP content, and the MMP levels in rat synovial tissues. The release of ROS increased significantly in MSU-induced AGA rats and decreased after RGFP966 and colchicine intervention (Fig 6A). After MSU induction, the production of ATP decreased significantly but increased after RGFP966 and colchicine intervention (Fig 6B). JC-1 staining was used to observe the effect of RGFP966 on the MMP (aggregate/monomer JC-1 ratio) of the synovial tissues (Fig 6C). RGFP966 and colchicine also blocked the MMP loss upon MSU-induced rats. These results suggested that RGFP966 could reduce MSU-triggered mitochondrial damage.

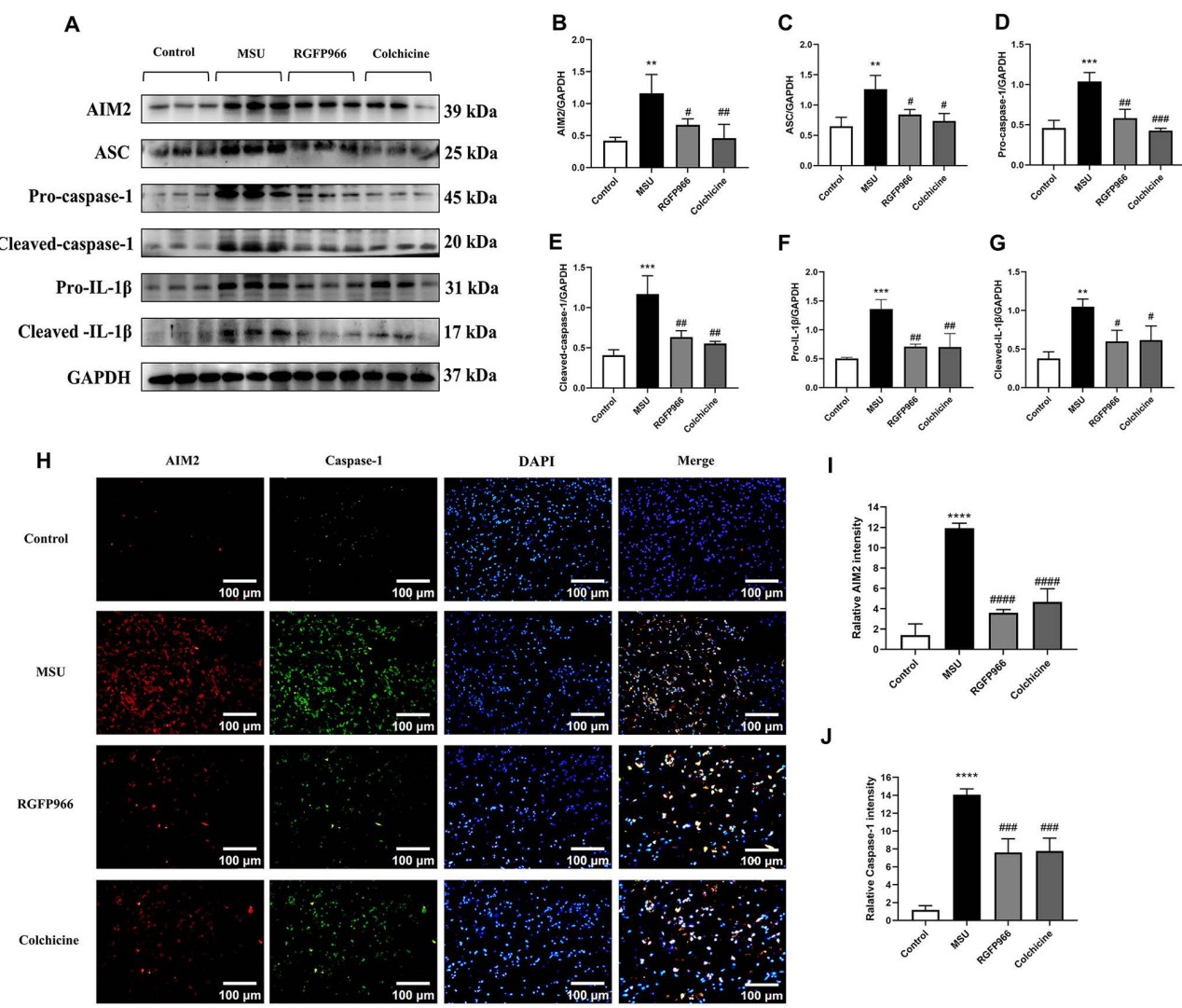

**Fig 5. RGFP966 decreased AIM2 inflammasomes expression in an experimental rat model of AGA.** Detection of protein expressions in the synovial tissues by Western blot (A). Statistical analysis of AIM2 (B), ASC (C), Pro-caspase-1 (D), Cleaved-caspase-1 (E), Pro-IL-1β (F) and Cleaved-IL-1β (G). AIM2 and Caspase-1 co-localization by immunofluorescence, showing fluorescence aggregates and inflammation after RGFP966 treatment (×200, Scale bar = 100 μm). AIM2 as red, and Caspase-1 as blue by immunofluorescent staining (H). The relative expression of AIM2 and Caspase-1 was lower in RGFP966 than in MSU (I, J). $^{**}P < 0.01$, $^{***}P < 0.001$ $^{****}P < 0.0001$ verus Control group; $^{#}P < 0.05$, $^{##}P < 0.01$, $^{###}P < 0.001$ verus MSU group. The data was presented as means ± standard deviation.

## RGFP966 promoted mitophagy in MSU-induced AGA rats

To investigate the role of mitophagy in AGA rats and the protective effect of RGFP966 in MSU-induced mitochondrial damage in AGA rats, Western blot and immunofluorescence staining were used to detect the expressions of mitophagy-related proteins. The results showed that the expression levels of Pink1 and Parkin were relatively low in the control group. In contrast, in the AGA model group, mitochondrial damage led to the activation of the Pink1/Parkin pathway, which facilitated the clearance of damaged mitochondria and initiated a certain level of mitophagy. Moreover, the expressions of Pink1, Parkin and the LC3-I to LC3-II conversion further significantly increased after RGFP966 and colchicine treatment (all $P < 0.05$) (Fig 7A-D). Immunofluorescence detection of Parkin and LC3 showed that only a few Parkin and LC3 spots

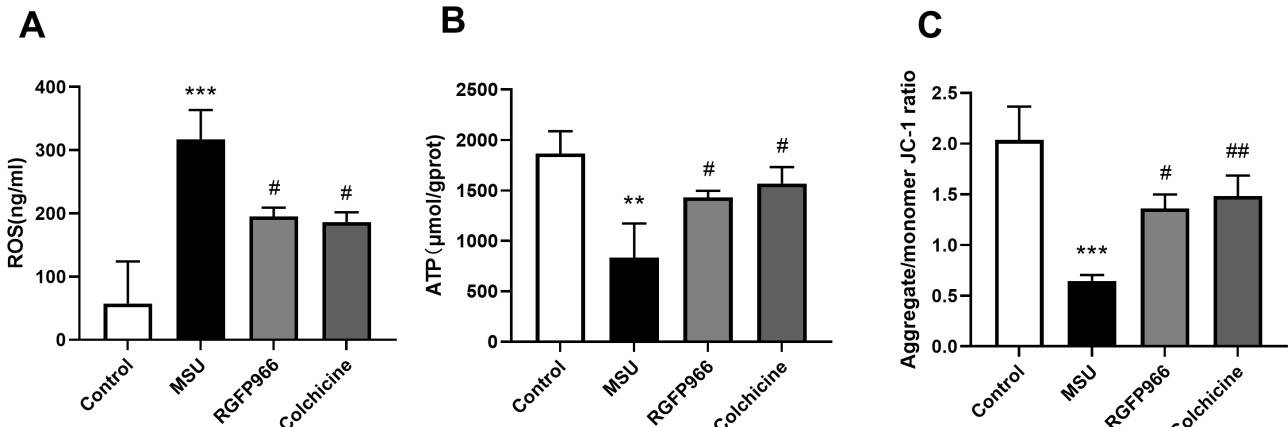

**Fig 6. RGFP966 decreased mitochondrial damage in AGA rats.** ROS content (A), ATP content (B), and MMP levels (C) were measured in synovial tissues. $^{**}P < 0.01$, $^{***}P < 0.001$ verus Control group; $^{#}P < 0.05$, $^{##}P < 0.01$ verus MSU group. The data was presented as means ± standard deviation.

were detected in the control group with no aggregation and almost no damage to the mitochondria and mitophagy, while in the MSU group, only a few aggregations of Parkin and LC3 spots were observed. The flake fluorescence with many aggregates was visible and increased significantly after RGFP966 and colchicine treatment (Fig 7E-G). These results suggested that RGFP966 partially inhibited mitochondrial damage by promoting mitophagy.

## Histopathologic observation of the kidneys in each group of rats

The HE staining results of the kidney tissues of rats in each group showed (Fig 8) that the glomeruli of the control rats were evenly distributed and the epithelial cells of the renal tubules were tightly arranged. In the MSU group, the glomeruli of rats showed atrophy or necrosis, and a large number of renal tubules were dilated. Compared with the MSU group, the necrosis of glomerular and tubular cells of rats in the RGFP966 group improved significantly, and the number and degree of tubular dilatation decreased significantly. However, the colchicine group did not show significant improvement compared with the MSU group, and the renal tubular epithelial cells showed vacuole-like degeneration.

## Discussion

Gout is a painful and complex type of arthritis characterized by the deposition of MSU crystals in the joints and subcutaneous tissues [32,33]. In our study, we established an animal model of AGA by injecting MSU into rats. Excessive neutrophil infiltration in the synovial tissues can cause severe pain and swelling [34]. The results of histological analysis showed that the synovial tissues of rats were structurally disorganized, with infiltration of inflammatory cells into the synovium. There was an increase on the swelling of the knee joints after MSU injection. It has been shown that MSU crystals upregulate the secretion of IL-1β, IL-18, IL-6 and TNF-α when interacting with monocytes, which is consistent with the results of this study [35,36]. While RGFP966 effectively decreased the joint swelling to almost control levels, and the gait score and synovitis score of the synovial tissues in rats were improved. The levels of inflammatory factors in serum were reduced, which effect is comparable with colchicine, suggesting that RGFP966 also had an appreciable anti-inflammatory effect.

Numerous studies have shown that the inflammatory response caused by MSU accumulation at the joint is associated with the NLRP3 inflammasome pathway [37,38]. AIM2 is an inflammasome distinct from the NLR pathway, and previous studies have found it to be involved in infection, autoinflammation and autoimmune disease, including rheumatoid arthritis (RA) and sjogren's syndrome (SS) [39,40]. Meanwhile, the AIM2 inflammasomes-mediated plays an important role in

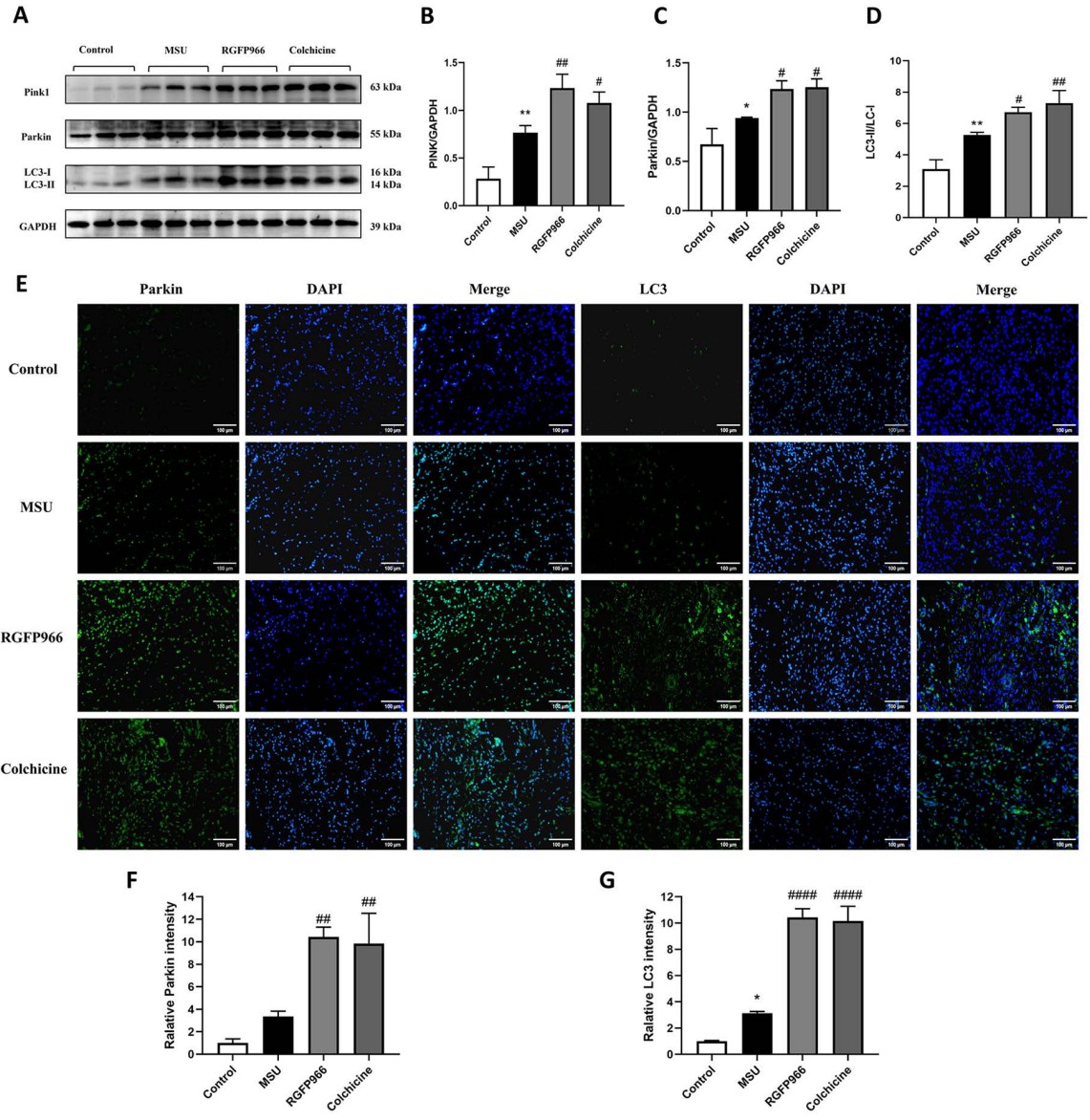

**Fig 7. RGFP966 promoted mitophagy in MSU-induced AGA rats.** Detection of protein expressions in synovial tissues by Western Blot (A). Statistical analysis of PINK1 (B), Parkin (C) and LC3 (D). RGFP966 treatment promoted the levels of Parkin and LC3 in synovial tissues, which was detected by immunofluorescence (×200, Scale bar = 100 μm) (E-G). $^*P < 0.05$, $^{**}P < 0.01$ verus Control group; $^#P < 0.05$, $^{##}P < 0.01$, $^{####}P < 0.0001$ verus MSU group. The data was presented as means ± standard deviation.

the development of AGA [7]. DNA, the main component of NETs, binds to the AIM2 inflammasomes in the cytoplasm of synovial cells in the joints, converting pro-caspase-1 to active Caspase-1 [41]. Cleaved Caspase-1 activates pro-IL-1β and pro-IL-18 into mature IL-1β and IL-18. IL-1β and IL-18 are key members of the IL-1 family, which are released extra-cellularly to recruit inflammatory cells to aggregate and amplify the inflammatory response through IL-1 receptors. Tian et al. have reported that MSU induces the release of dsDNA into the cytosol and activates the AIM2 inflammasomes through certain pathways, thereby regulating the cleavage and activation of Caspase-1 [42,43]. In our study, Western blot and immunofluorescence showed that the activation of the level of AIM2 and Cleaved-caspase-1 were increased

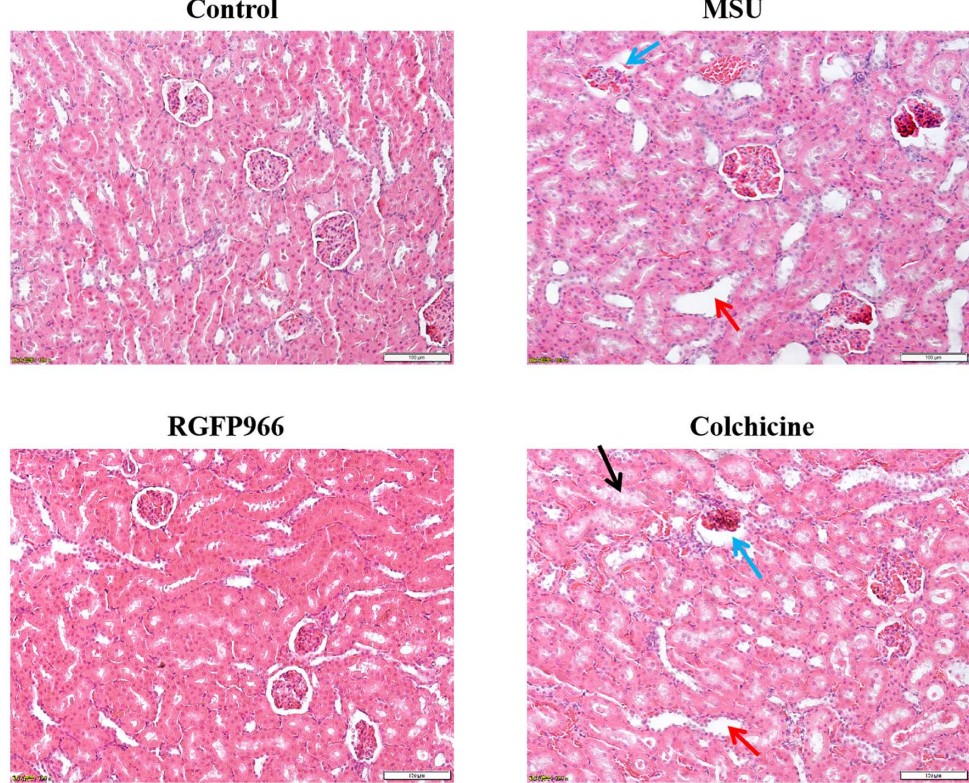

**Fig 8. Pathological changes of rat kidney tissues (×200, Scale bar = 100 μm).** Blue arrow: glomeruli atrophy; red arrow: renal tubules dilatation; black arrow: vacuole-like degeneration.

significantly in AGA rats, and Cleaved-caspase-1 levels were positively correlated with AIM2 protein levels. It suggested that the expression of Caspase-1 may be related to the activation of AIM2 inflammasomes, which are collectively involved in the development of AGA. The level of Caspase-1 decreased significantly after RGFP966 treatment, suggesting that the intervention of RGFP966 in the development of AGA is related to the inhibition of the activation of AIM2 signaling and the reduction of inflammatory response.

Yu et al. have reported that AIM2 triggers the activation of Caspase-1 to accelerate the mitochondrial catabolism by various pathways, resulting in the generation of ROS, dissipation of the MMP and disruption of the mitochondrial network [24]. Typically, a decrease in ATP levels also indicates damaged or decreased mitochondrial function, which usually occurs in conjunction with a decrease in the MMP. The JC-1 staining method was used in this study to detect the MMP [44,45]. MSU-induced AGA had a low MMP, which increased significantly after RGFP966 treatment. Mitochondria in synovial tissues released more ROS and produced less ATP after MSU induction, and RGFP966 reversed the above changes. These results suggested that RGFP966 ameliorated the damaged mitochondrial dysfunction in AGA rats.

Mitophagy, a selective type of cellular autophagy, maintains mitochondrial function by removing damaged mitochondria and reducing excessive accumulation of ROS [46]. Pink1 is a type of serine/threonine protein kinase, which mainly exists in the inner mitochondrial membrane under normal conditions. Parkin, an E3 ubiquitin ligase, which was first identified in Parkinson's disease, both of them have a synergistic effect on each other, and together they contribute to the activation of mitophagy [47,48]. Parkin, upon activation, further induces ubiquitination of

 

various mitochondrial outer membrane proteins. The ubiquitinated proteins can bind to p62, a ubiquitin-binding autophagy receptor and signaling protein which further interacts with LC3 and then recruits autophagic vesicles to stimulate mitophagy to phagocytose damaged mitochondria [49]. During the clearance of damaged mitochondria, Pink1 accumulates in the damaged mitochondrial outer membrane, followed by the recruitment of Parkin in the cytoplasm to the mitochondria. Therefore, Pink1/Parkin has also been considered as a protective protein that avoids excessive production of ROS and may also function to ameliorate the inflammatory response. Numerous experimental results indicate that mitophagy is involved in the pathogenesis of gout, and that it can inhibit and delay the development of gout by promoting the occurrence of mitophagy. Caspase-1 triggers mitochondrial damage upon its activation by the AIM2 inflammasomes, inhibiting mitophagy to amplify mitochondrial damage, mediated in part by cleavage of the key mitophagy regulator Parkin [26]. In the present study, to further investigate whether RGFP966 regulates mitophagy by inhibiting activation of AIM2 inflammasomes to relieve AGA, we detected mitophagy-related proteins. In the AGA model, the levels of Pink1, Parkin and LC3-II/LC-I were elevated, suggesting that the organism initiates a certain degree of mitophagy to inhibit the damage caused by AGA. On the other hand, Zhong et al. have reported that mitophagy may generated anti-inflammatory mechanism by restraining excessive IL-1β and IL-18 production [50]. Furthermore, mitophagy inhibits the AIM2 inflammasomes activation by removing damaged mitochondria, regulating mitochondrial mass, and maintaining mitochondrial homeostasis. The current study showed that RGFP966 could further up-regulated the expressions of Pink1, Parkin and LC3-II/LC-I, which is consistent with the results of the previous findings of Tao et al. [51]. Therefore, these results showed that RGFP966 effectively alleviated AGA by promoting the Pink1/Parkin signaling pathway to promote mitophagy.

According to the 2020 ACR gout treatment guideline suggested colchicine and non-steroid anti-inflammatory drugs are the common drugs for AGA [52]. Although they can suppress inflammation and improve pain relatively quickly, it has more adverse effects, including gastrointestinal bleeding, liver or kidney damage [53,54]. Most of the natural medicines for AGA are administered orally, and intra-articular injection is a common clinical method used in the treatment of bone and joint diseases, which can be used to alleviate joint swelling and pain and injury [55,56]. In the study, RGFP966 was chosen as an intra-articular injection drug for AGA, and the therapeutic effect on AGA and its possible mechanisms were investigated. The main advantage of intra-articular drug injection is that the drug leaks from the joint cavity and rapidly enters the systemic circulation, thus prolonging the residence time of the drug in the joint cavity. This method has a maximum effect with the minimum dose and reduces the systemic side effects [57,58]. In this study, the body weight of rats after colchicine treatment was reduced and the glomeruli in the kidneys were atrophied, suggesting that it had some adverse effects, which is consistent with the results of the previous findings of Chen et al. [59]. The morphologic structures of the kidney tissues of the rats were intact and the boundaries were obvious, which indicated that RGFP966 did not have any significant effects on the kidneys. Therefore, RGFP966 offers certain safety advantages over colchicine in the treatment of AGA. In summary, the results obtained clearly indicated that RGFP966 could probably inhibit the activation of the AIM2 inflammasomes to promote mitophagy in AGA rat models, which provided experimental evidence for the treatment of gout. However, for the studies of dose-dependent effects of RGFP966 on AGA rats, we only used one dose and studies with more doses are warranted to further clarify this finding. In the future, more investigations are needed to improve our understanding of RGFP966.

## Conclusion

In summary, RGFP966 could probably inhibit the activation of the AIM2 inflammasomes to promote mitophagy through the Pink1/Parkin signaling pathway, thereby relieving AGA (Fig 9). Collectively, our findings may provide a novel avenue for therapeutic strategies against AGA in the future.

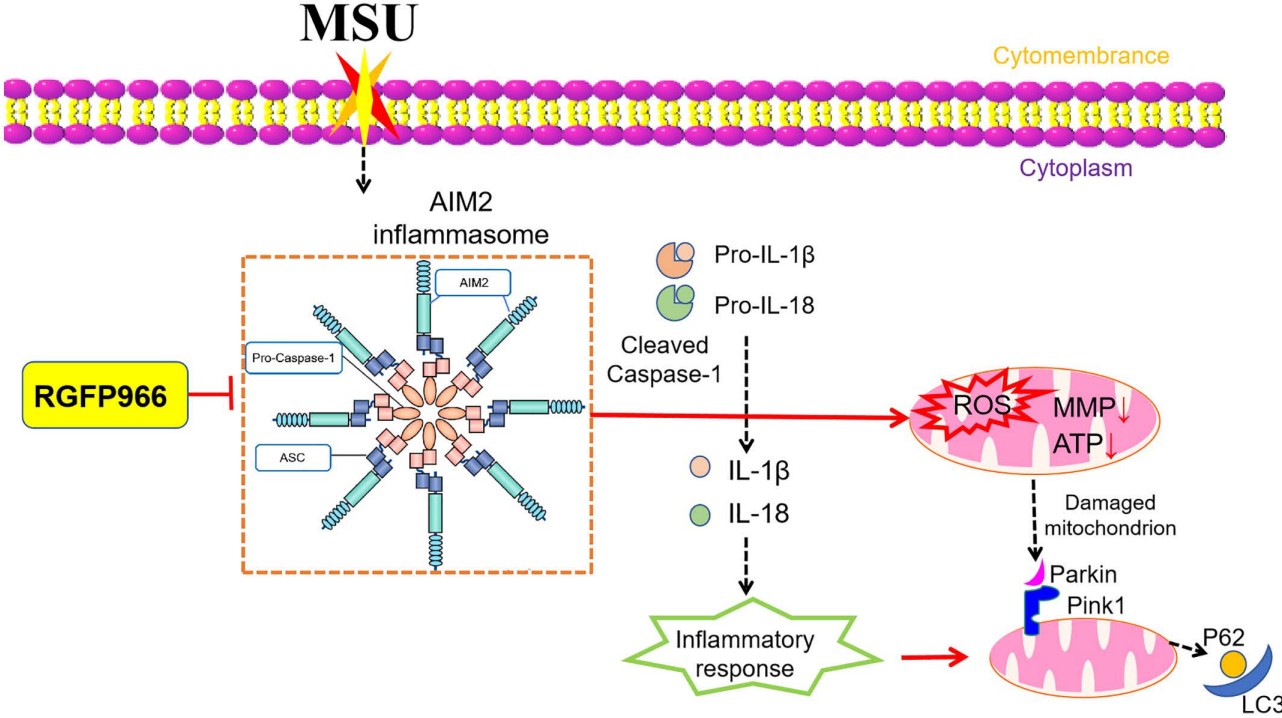

**Fig 9. Mechanism of RGFP966-promoted mitophagy via inhibiting the activation of the AIM2 inflammasomes.** MSU induces joint inflammation by activating the AIM2 inflammasomes, promoting the secretion of IL-1β and IL-18, and causing mitochondrial damage. RGFP966 alleviates AGA by inhibiting the activation of the AIM2 inflammasomes and promoting mitophagy to clear damaged mitochondria.

## Supporting information

**S1 File. Raw images.**
(PDF)

**S2 File. Data.**
(ZIP)

## Author contributions

**Conceptualization:** Zewei Wang, Xiuyun Shi, Hongxu Jin, Ying Liu.

**Data curation:** Xiuyun Shi, Jing Dai.

**Formal analysis:** Chunyang Liu.

**Funding acquisition:** Jing Tian.

**Investigation:** Zewei Wang, Jing Dai.

**Methodology:** Zewei Wang.

**Project administration:** Hongxu Jin, Ying Liu.

**Resources:** Feng Cheng, Xiuyun Shi, Hongxu Jin, Ying Liu.

**Software:** Zewei Wang.

**Supervision:** Chunyang Liu, Jing Tian.

**Validation:** Feng Cheng, Xiuyun Shi, Jing Dai.

**Visualization:** Feng Cheng, Xiuyun Shi.

**Writing – original draft:** Zewei Wang.

**Writing – review & editing:** Chunyang Liu, Feng Cheng, Xiuyun Shi, Jing Dai, Jing Tian, Hongxu Jin, Ying Liu.

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
