## [Decision Letter · Decision Letter 0]

18 Feb 2026

PONE-D-25-67001RGFP966 inhibits activation of AIM2 inflammasomes to promote mitophagy to relieve acute gouty arthritisPLOS One

Dear Dr. Liu,

Thank you for submitting your manuscript to PLOS ONE. After careful consideration, we feel that it has merit but does not fully meet PLOS ONE’s publication criteria as it currently stands. Therefore, we invite you to submit a revised version of the manuscript that addresses the points raised during the review process.

We look forward to receiving your revised manuscript.

Kind regards,

David Chau

Academic Editor

PLOS One

Journal Requirements:

4. To comply with PLOS One submissions requirements, in your Methods section, please provide additional information regarding the experiments involving animals and ensure you have included details on (1) methods of sacrifice, (2) methods of anesthesia and/or analgesia, and (3) efforts to alleviate suffering.

5. Please provide a complete Data Availability Statement in the submission form, ensuring you include all necessary access information or a reason for why you are unable to make your data freely accessible. If your research concerns only data provided within your submission, please write "All data are in the manuscript and/or supporting information files" as your Data Availability Statement.

7. Thank you for stating the following financial disclosure:

Natural Science Foundation of Liaoning Province 2023-MSLH-361

Reviewers' comments:

Reviewer's Responses to Questions

**Comments to the Author**

1. Is the manuscript technically sound, and do the data support the conclusions?

Reviewer #1: Partly

Reviewer #2: Partly

Reviewer #3: Partly

2. Has the statistical analysis been performed appropriately and rigorously? 

Reviewer #1: Yes

Reviewer #2: Yes

Reviewer #3: Yes

3. Have the authors made all data underlying the findings in their manuscript fully available?

Reviewer #1: Yes

Reviewer #2: Yes

Reviewer #3: Yes

4. Is the manuscript presented in an intelligible fashion and written in standard English?

Reviewer #1: No

Reviewer #2: Yes

Reviewer #3: No

5. Review Comments to the Author

Reviewer #1: The authors should clarify their findings. They should shorten their discussion, and their results. They should compare their findings with those of other authors within the discussion. The manuscript should have some figures depicting the results.

Reviewer #2: Your study addresses an important and clinically relevant question in gout—whether the degree of hyperuricemia tracks with BMD and remodeling markers and you provide consistent directionality across correlation, multivariable linear regression, and spline modeling. The strongest revisions would be to ensure outcome classification is fully appropriate for the enrolled age range, tighten analytic transparency/model stability for the binary outcome, and harmonize effect-size language (absolute vs percent) so results are readily interpretable

1. A key point is that the manuscript applies WHO T-score categories (osteoporosis ≤−2.5; osteopenia −1 to −2.5) to a cohort enrolled from 18–80 years, and then combines osteoporosis (n=15) with osteopenia into a single “low BMD” composite. This is important because the clinical interpretation and labeling of “osteopenia/osteoporosis” can differ in younger adults, and categorization choices can influence prevalence estimates and downstream regression results. It would be helpful to clarify which guideline framework was intended for participants <50 years and to strengthen robustness by repeating analyses using an age-appropriate classification (and/or a sensitivity analysis restricted to age ≥50).

2. The study is restricted to hospitalized primary gout patients, excludes recent flares and any urate-lowering or urate-affecting medications in the prior 3 months, and notes this may best represent “stable gout.” The flow diagram also indicates substantial attrition before the final analytic sample (patients lacking BMD/BTMs). This is consequential because the observed associations may reflect a selected subset with specific care pathways, comorbidity patterns, or disease phenotype, which can shift effect sizes and subgroup patterns relative to community/outpatient gout. I suggest expanding the characterization of included vs excluded patients (if available) and calibrating external claims to emphasize the inpatient/stable-gout context.

3. The manuscript appropriately notes that cross-sectional data cannot establish causality/temporality and acknowledges missing data on chronic inflammation, diet, and physical activity. At the same time, the Results/Conclusions emphasize “independent risk factor” language and highlight a threshold-like interpretation. This is a concern because serum urate may be acting as a marker of gout phenotype or other unmeasured exposures that also influence BMD (e.g., tophus burden, disease course), and such residual confounding can meaningfully affect “independence” claims. It would strengthen interpretability to reframe conclusions toward association, clarify plausible causal directions, and add sensitivity analyses using available proxies (e.g., tophus status/duration) and/or excluding sparsely represented extremes.

4.The manuscript models a binary “osteopenia” outcome (osteopenia+osteoporosis composite) with 79 low-BMD cases, while also presenting a broad set of candidate predictors (anthropometrics, labs, comorbidities, tophi). This is important to address because limited event counts can yield unstable multivariable ORs, inflate apparent “independence,” and increase sensitivity to modeling choices or collinearity (notably BMI vs WC/HC). I recommend explicitly documenting the model-building approach (pre-specification vs selection), reporting collinearity checks (e.g., VIF), and providing internal validation or shrinkage (bootstrap or penalized regression) so the serum urate signal can be judged for robustness.

5. The restricted cubic spline analysis is described as linear (nonlinearity p>0.05), but the Results/Mini Abstract highlight “sUA levels >9.40 mg/dL” as if it were a clinically validated threshold. This deserves tightening because readers may infer a data-supported cutpoint when it may instead reflect the plotting reference or where the modeled OR crosses 1 in this sample, and the manuscript itself notes reduced precision at extremes due to sparse data. I propose clearly defining how 9.40 mg/dL was derived (reference vs data-driven crossing), presenting predicted ORs at clinically meaningful percentiles, and rephrasing toward a continuous gradient unless the cutpoint is validated.

6. Table 2 reports “Age (years) r = 0.98, P = 0.089” for lumbar spine BMD; this value looks internally inconsistent with the stated P-value and likely needs formatting/decimal correction.

7. Table 2 has “WC (cm) 0.011* 0.849” where the asterisk appears attached to the correlation coefficient despite a non-significant P.

8.Logistic regression narrative states BMI OR=0.44 and sUA CI lower bound 1.62, while Figure 2 shows different values (e.g., BMI OR≈0.36; sUA CI lower bound ≈1.46).

9.Quartile bounds are inconsistent: Results define Q2 as “>8.49–9.36” while Table 3 lists Q2 as “8.47–9.36.”

Reviewer #3: The concept of the study is good and the experimental design is strong; however, the manuscript would benefit from improved organization, as well as corrections to English and grammar.

Introduction: While AIM2 activation is described, AGA is primarily associated with NLRP3 inflammasome activation. The manuscript does not justify the emphasis on AIM2 over NLRP3. Additionally, the final paragraph should clearly state the study objective and hypothesis; methodological details are better suited for the Methods section.

Methods (Line 95): The group descriptions are unclear. Please specify whether Group 3 represents the RGFP966-only treatment group or RGFP966 combined with MSU.

Line 134: The word “cuted” is incorrect and should be replaced with “cut.”

Line 135 (mitochondrial extraction method): Providing the centrifugation speed as an RCF value would improve reproducibility.

6. PLOS authors have the option to publish the peer review history of their article (what does this mean? ). If published, this will include your full peer review and any attached files.

**Do you want your identity to be public for this peer review?** For information about this choice, including consent withdrawal, please see our Privacy Policy .

Reviewer #1: No

Reviewer #2: No

Reviewer #3: No

---

## [Author Response · Author response to Decision Letter 1]

6 Mar 2026

Dear editor and reviewer,

Thank for your professional review on my article. As you are concerned, there are some problems that need to be addressed. According to your nice suggestions, we have made extensive corrections to our previous draft, and the changes are highlighted in the manucript. Our point-to-point responses are as follows.

Reviewer#1:

The authors should clarify their findings. They should shorten their discussion, and their results. They should compare their findings with those of other authors within the discussion. The manuscript should have some figures depicting the results.

Response: Thanks for your positive comments. Based on your comments, we have shorten discussion and results. We have added some figures depicting the results in the Results section in the revised research. Some findings have listed compared with our findings in the Discussion section.

Reviewer#2:

We sincerely thank the reviewer for careful reading. However, this reviewer's comments are unrelated to my manuscript.

Reviewer#3:

1.Introduction: While AIM2 activation is described, AGA is primarily associated with NLRP3 inflammasome activation. The manuscript does not justify the emphasis on AIM2 over NLRP3. Additionally, the final paragraph should clearly state the study objective and hypothesis; methodological details are better suited for the Methods section.

Response: We would like to express our sincere gratitude for editor’ comments and valuable feedback. The revised content of the paper is as follows: Inflammasomes, composed of pattern recognition receptors (PRRs), apoptosis-associated speck-like protein containing an apoptosis-associated speck-like protein (ASC), and inactive pro-caspase-1, are multiprotein complexes found in the cytoplasm [6]. They can only function after they are activated. The PRRs of inflammasomes mainly include members of the nucleotide-binding oligomerization domain-like receptor (NLR) family and the absent in melanoma 2 (AIM2)-like receptor (ALR) family [7]. (Page 3)

References：

6. Lu A, Magupalli VG, Ruan J, Yin Q, Atianand MK, Vos MR, et al. Unified polymerization mechanism for the assembly of ASC-dependent inflammasomes. Cell.2014;156:1193–206. doi: 10.1016/j.cell.2014.02.008

7. Chu J, Tian J, Li P, Fu D, Guo L, Sun R. The impact of AIM2 inflammasome-induced pyroptosis on acute gouty arthritis and asymptomatic hyperuricemia patients. Front Immunol. 2024;15:1386939. doi: 10.3389/fimmu.2024.1386939.

In order to reveal the role and mechanism of RGFP966-mediated AIM2 inflammasomes in regulation of Pink1/Parkin signaling pathway, this study established an AGA rat model and used colchicine as the positive control. We investigated the effects of RGFP966, a potential AIM2 inhibitor, on inflammatory and mitophagy-related markers in rat synovial tissues, thereby providing new insights into potential therapeutic targets for gout prevention and treatment. (Page 5)

2.The group descriptions are unclear. Please specify whether Group 3 represents the RGFP966-only treatment group or RGFP966 combined with MSU.

Response: Considering the editors’s suggestion, we have corrected into RGFP966+MSU and Colchicine+MSU in the Experiment design section on page 5.

3. The wordcuted” is incorrect and should be replaced with “cut.”

Response: We sincerely thank the reviewer for careful reading. As suggested by the reviewer, we have corrected into “cut”.

4.Providing the centrifugation speed as an RCF value would improve reproducibility.

Response: We sincerely thank the reviewer for careful reading. As suggested by the reviewer, we have provided the centrifugation speed on page 7.

---

## [Decision Letter · Decision Letter 1]

31 Mar 2026

PONE-D-25-67001R1RGFP966 inhibits activation of AIM2 inflammasomes to promote mitophagy to relieve acute gouty arthritisPLOS One

Dear Dr. Liu,

Thank you for submitting your manuscript to PLOS ONE. After careful consideration, we feel that it has merit but does not fully meet PLOS ONE’s publication criteria as it currently stands. Therefore, we invite you to submit a revised version of the manuscript that addresses the points raised during the review process.

We look forward to receiving your revised manuscript.

Kind regards,

David Chau

Academic Editor

PLOS One

Journal Requirements:

Reviewers' comments:

Reviewer's Responses to Questions

**Comments to the Author**

1. If the authors have adequately addressed your comments raised in a previous round of review and you feel that this manuscript is now acceptable for publication, you may indicate that here to bypass the “Comments to the Author” section, enter your conflict of interest statement in the “Confidential to Editor” section, and submit your "Accept" recommendation.

Reviewer #1: All comments have been addressed

Reviewer #2: All comments have been addressed

Reviewer #3: All comments have been addressed

2. Is the manuscript technically sound, and do the data support the conclusions?

Reviewer #1: Yes

Reviewer #2: Yes

Reviewer #3: Yes

3. Has the statistical analysis been performed appropriately and rigorously? 

Reviewer #1: Yes

Reviewer #2: Yes

Reviewer #3: Yes

4. Have the authors made all data underlying the findings in their manuscript fully available?

Reviewer #1: Yes

Reviewer #2: Yes

Reviewer #3: Yes

5. Is the manuscript presented in an intelligible fashion and written in standard English?

Reviewer #1: No

Reviewer #2: Yes

Reviewer #3: Yes

6. Review Comments to the Author

Reviewer #1: The authors should improve the use of the English language. The use of the English language should be improved throuhout the manuscript including the abstract, the introduction, the methods section, the results section and of course the discussion and the conclusions section. If the use of the English language is improved it will be easier for the readers to comprehend the meaning of the paper and to understand the aim and significance of the experiments which are described in sufficient detail. Improvement of the use of the English language will make the paper understandable to a large audience all over the world.

Reviewer #2: The manuscript is good enough now and i appreciate all your efforts to address it properly and I have no additional comments

Reviewer #3: (No Response)

7. PLOS authors have the option to publish the peer review history of their article (what does this mean? ). If published, this will include your full peer review and any attached files.

**Do you want your identity to be public for this peer review?** For information about this choice, including consent withdrawal, please see our Privacy Policy .

Reviewer #1: No

Reviewer #2: No

Reviewer #3: No

---

## [Author Response · Author response to Decision Letter 2]

12 Apr 2026

Dear editor and reviewer,

Thank for your professional review on my article. As you are concerned, there are some problems that need to be addressed. According to your nice suggestions, we have made extensive corrections to our previous draft, and the changes are highlighted in the manucript. Our point-to-point responses are as follows.

Reviewer#1:

Reviewer #1: The authors should improve the use of the English language. The use of the English language should be improved throuhout the manuscript including the abstract, the introduction, the methods section, the results section and of course the discussion and the conclusions section. If the use of the English language is improved it will be easier for the readers to comprehend the meaning of the paper and to understand the aim and significance of the experiments which are described in sufficient detail. Improvement of the use of the English language will make the paper understandable to a large audience all over the world.

Response: Thanks for your positive comments. Based on your comments, we tried our best to improve the manuscript and polish the language. These changes will not influence the content and framework of the paper. And here we did not list the changes but marked in red in the revised paper. We appreciate for Editors/Reviewers’ warm work earnestly and hope that the correction will meet with approval.

---

## [Decision Letter · Decision Letter 2]

14 Apr 2026

RGFP966 inhibits activation of AIM2 inflammasomes to promote mitophagy to relieve acute gouty arthritis

PONE-D-25-67001R2

Dear Dr. Liu,

We’re pleased to inform you that your manuscript has been judged scientifically suitable for publication and will be formally accepted for publication once it meets all outstanding technical requirements.

Kind regards,

David Chau

Academic Editor

PLOS One

Additional Editor Comments (optional):

Reviewers' comments:

Reviewer's Responses to Questions

**Comments to the Author**

1. If the authors have adequately addressed your comments raised in a previous round of review and you feel that this manuscript is now acceptable for publication, you may indicate that here to bypass the “Comments to the Author” section, enter your conflict of interest statement in the “Confidential to Editor” section, and submit your "Accept" recommendation.

Reviewer #1: All comments have been addressed

2. Is the manuscript technically sound, and do the data support the conclusions?

Reviewer #1: (No Response)

3. Has the statistical analysis been performed appropriately and rigorously? 

Reviewer #1: Yes

4. Have the authors made all data underlying the findings in their manuscript fully available?

Reviewer #1: Yes

5. Is the manuscript presented in an intelligible fashion and written in standard English?

Reviewer #1: Yes

6. Review Comments to the Author

Reviewer #1: The authors have made the necessary amendments. The manuscript meets the requirements for publication in Plos One. It may now be published.

7. PLOS authors have the option to publish the peer review history of their article (what does this mean? ). If published, this will include your full peer review and any attached files.

**Do you want your identity to be public for this peer review?** For information about this choice, including consent withdrawal, please see our Privacy Policy .

Reviewer #1: No

---

## [Editor Report · Acceptance letter]

PONE-D-25-67001R2

PLOS One

Dear Dr. Liu,

I'm pleased to inform you that your manuscript has been deemed suitable for publication in PLOS One. Congratulations! Your manuscript is now being handed over to our production team.

Kind regards,

on behalf of

Dr. David Chau

Academic Editor

PLOS One